# Development and Investigation of PEDOT:PSS Composition Coated Fabrics Intended for Microwave Shielding and Absorption

**DOI:** 10.3390/polym13081191

**Published:** 2021-04-07

**Authors:** Vitalija Rubeziene, Julija Baltusnikaite-Guzaitiene, Ausra Abraitiene, Audrone Sankauskaite, Paulius Ragulis, Gilda Santos, Juana Pimenta

**Affiliations:** 1Department of Textiles Physical-Chemical Testing, Center for Physical Sciences and Technology, 48485 Kaunas, Lithuania; vitalija.rubeziene@ftmc.lt; 2Department of Textile Technologies, Center for Physical Sciences and Technology, 48485 Kaunas, Lithuania; ausra.abraitiene@ftmc.lt (A.A.); audrone.sankauskaite@ftmc.lt (A.S.); 3Microwave Laboratory of Physical Technology Department, Center for Physical Sciences and Technology, 10257 Vilnius, Lithuania; paulius.ragulis@ftmc.lt; 4CITEVE-Technological Centre for the Textile and Clothing Industries of Portugal, 4760-034 Vila Nova de Famalicão, Portugal; gsantos@citeve.pt; 5CeNTI-Centre for Nanotechnology and Smart Materials, 4760-034 Vila Nova de Famalicão, Portugal; jpimenta@centi.pt

**Keywords:** conductive textiles, EMR shielding, plasma treatment, microwave absorbing material

## Abstract

This study presents the investigation of the electromagnetic properties and resistance performance of electrically conductive fabrics coated with composition containing the conjugated polymer system poly(3,4-ethylenedioxythiophene)-polystyrene sulfonate (PEDOT:PSS). The developed fabrics were intended for electromagnetic radiation (EMR) shielding in microwave range and for absorbing microwaves in radar operating range, so as to act as radar absorbing materials (RAM). The measurements of reflection and transmission of the developed fabrics were performed in a frequency range of 2–18 GHz, which covers the defined frequencies relevant to the application. Four types of fabrics with different fiber composition (polyamide; polyamide/cotton; wool and para-aramid/viscose) were selected and coated with conductive paste using screen printing method. It was found that EMR shielding effectiveness (SE) as well as absorption properties depend not only the amount of conductive paste topped on the fabric, but also resides in the construction parameters of fabrics. Depending on such fabric structural parameters as density, mass per unit area, type of weave, a layer of shield (or coating) just sticks on the fabric surface or penetrates into fabric, changing the shield thickness and herewith turning SE results. Meanwhile, the fiber composition of fabrics influences mostly bonding between fibers and polymer coating. To improve the resistance performance of the developed samples, a conventional textile surface modification technique, atmospheric plasma treatment, was applied. Initially, before plasma treatment and after treatment the fabrics were evaluated regarding an aqueous liquid repellency test, measuring the contact angles for the water solvent. The influence of plasma treatment on resistance performance of coated fabrics was evaluated by subjecting the plasma treated samples and untreated samples to abrasion in the Martindale abrasion apparatus and to multiplex washing cycles. These investigations revealed that applied plasma treatment visibly improved abrasion resistance as a result of better adhesion of the coating. However, washing resistance increased not so considerably.

## 1. Introduction

The reduction of electromagnetic radiation (EMR) impact is very important for the protection of people frequently using electrical and electronic devices which can emit electromagnetic waves with frequencies that are potential hazards to health. The International Agency for Research on Cancer (IARC) based on literature reports classifies radio frequency electromagnetic fields as group 2B, which includes factors probably carcinogenic to human [1]. The most utilized range is the microwave range, which can be defined as 1 −40 GHz, as most of the modern point to point, wireless, and satellite communications occupy this range. Electrically conductive woven or knitted fabrics with particular EMR shielding properties not only offer an opportunity to counter these threats, but also can be applicable to develop radar absorbing materials (RAM), for use in the field of stealth technology to disguise a vehicle or soldier from radar detection [2].

EMR shielding within the given frequency range can be provided by corresponding EMR reflection or absorption, and in the best case by reflection and absorption at the same time. In order to absorb or reflect EMR, materials must interact with either the electric or magnetic field of the radiation. Textile materials with incorporated conductive additives or coated with special conductive formulations are electrically conductive and therefore interact with the electric component of EMR [3].

The desired property of EMR shielding textile materials is low transmission that means high shielding effectiveness—SE (dB). According to the requirements of EMR shielding textiles on general use [4], conductive textiles can be classified in five grades from a fair grade to an excellent one: fair—7 dB ≥ SE ˃ 5 dB; moderate—10 dB ≥ SE ˃ 7 dB; good—20 dB ≥ SE ˃ 10 dB; very good—30 dB ≥ SE ˃ 20 dB; excellent—SE ˃ 30 dB. For EMR shielding applications, typically SE of at least 20 dB (indicates that 99% of the electromagnetic energy is reflected or absorbed by the material) is needed. SE of 30 dB indicates that 99.9% of the EM energy is reflected or absorbed by the material, with only 0.1% exiting the shielding material [5].

However, among EMR shielding textile materials only materials with substantial contribution to shielding from absorption have the potential to be used as radar absorbing materials (RAM). It is stated [6] that, depending on their practical use, such materials should be characterised by a high EMR absorption coefficient, even twice as high as their reflection coefficient within as wide a frequency band as possible. During the investigation of EMR shielding characteristics of the textile fabrics with different deposit of conductive additives [7] it was found that absorption dominated when the total EMR SE was below 20 dB and that reflectance dominated when EMR SE was above 20 dB. Hence, in order to be effective for a radar signature reduction application, SE must be not too high since such a textile material would be too reflective, resulting in the poor radar protection properties [8,9].

There are various techniques to provide the textiles with electrical conductivity and provide them with electromagnetic properties: introduction of electrically conductive yarns (carbon fibres, metal fibre); metallization of fabrics or yarns (voltaic, vacuum vaporisation); lamination or coating of conductive layers onto the fabric surface with metal particles, transparent organic metal oxides, carbon or inherently conducting polymers (ICPs).

Variable electrical conductivity, electromagnetic shielding, electrostatic properties, and in comparison, the low cost of ICPs, have led to the investigation of potential applications of these materials not only as corrosion protectors, sensors, polymer actuators, but also as electromagnetic shields and radar absorbers. In contrast to some commonly used metallic shielding materials, conducting polymers not only reflect but also absorb EMR in the microwave frequency range [10,11]. The dominant shielding characteristic of absorption other than that of reflection for metals render ICPs more promising materials in applications requiring not only high EMR shielding effectiveness but also shielding by absorption, such as in stealth technology.

The studies of textiles coated with conductive polymers [12,13] show that they are not highly effective as EMR shielding materials owing to their medium-level conductivity and therefore large skin depth. Textile fabrics with ICPs coatings mostly demonstrate shielding effectiveness (SE) not exceeding 20 dB in the microwave frequency range. Combine with fact that coatings are naturally thin, they cannot act as effective reflective barriers to EMR radiation. However, because they are highly absorptive in the microwave region, microwave-absorbing composite materials can be designed in conjunction with textiles.

The past works concerning EMR shielding with ICPs on textile fabrics are focused mainly on polyaniline (PANI) and polypyrrole (PPY) applications [7,10,13,14,15,16]. However, increasingly appears publications about application of other ICPs-PEDOT, for development of EMR shielding textiles [8].

The conjugated polymer system—poly(3,4-ethylenedioxythiophene)-polystyrene sulfonate (PEDOT:PSS) was chosen for this study because of its processability, stable electrical conductivity and low price in comparison with other ICPs. Also, PEDOT:PSS has other merits. For example, hydrogel particles offer excellent processing properties for the production of thin, transparent, conducting films [17]; coating with PEDOT:PSS does not affect the mechanical properties of the substrate and allows for them to be used as flexible and deformable substrates [18]. Whereas, due to the black color of polyaniline or polypyrrole, these ICPs are unlikely for developing conductive camouflage materials as the colour of these materials in VIS range should remain unchanged after coating.

The most common techniques [8] for applying ICPs on textiles are solution coating [19], in-situ polymerization [20], polymerization in supercritical fluid, electrochemical polymerization [21], electrostatic spinning [22], coating by screen printing [23], inkjet printing [24], or knife-over-roll technology [25]. The electrical properties of the conducting polymer-coated fabrics are influenced by various factors such as the concentration of reactants, deposit and thickness of the polymer coating, nature of the substrate surface, binding strength of coating to the textile surface, etc. [25,26]. Coating usually does not change the flexibility of the fabrics if it is applied in very thin layer, low mass and closed fabric structures. Most of the commercially available EMR shielding fabrics are produced by coating technologies and have very homogeneous and closed structures thus exhibiting satisfactory EMR shielding capabilities and isotropic behaviour [27].

Requirements for ideal EMR shielding textile or RAM textile fabric are not only high enough relevant EMR shielding effectiveness or absorption, over wide frequency range, but also stable electrical properties, resistance to washing, withstanding other impacts, which appear during the wear. However, current coating technologies do not provide sufficiently substantial bonding between textile substrate and ICPs layer. That leads to loss of conductivity of coated textile, caused by poor adhesion, during wear or after washing procedures. So, the resistance performance of fabrics coated with ICPs polymers needs to be improved. Most of the conductive coatings under studies are obtained by solution coating or in-situ polymerization and most commonly the materials so produced are used in development of electronic textiles, sensor [28,29]. However, the research into the development of the RAM textile and especially applying coating by screen printing or knife-over-roll technology, which allows the finish to be applied only on one surface/side of the material investigated, is not analyzed.

Plasma treatment is one of a several methods to improve bonding between the substrate and any polymeric coating. Plasma not only changes the surface morphology of the substrate but also binds active sites to the surface, rendering the surface active for subsequent reactions [30]. Plasma treatment is a useful method to introduce functional radicals onto the surfaces of material without changing material’s bulk properties [31]. Plasma technology has the advantages of convenience and environmental friendliness and more importantly, it has the capacity to activate the surface of substrates [32,33]. During plasma processing, many electrons, ions and metastable ions in the ion flow can break the molecular chain on the surface in a short time, increase the number of active groups and unsaturated bonds, and meanwhile play a certain etching effect [34]. Plasma treatment occurred only on the surface without affecting the elemental composition and macroscopic mechanical properties of the substrate [35]. Therefore, low-temperature plasma treatment could be employed to treat substrates before coating with conductive pastes.

The aim of this work was to develop and investigate the fabrics coated with formulation containing PEDOT:PSS, focusing on their electromagnetic properties and resistance performance. The developed fabrics are intended for electromagnetic radiation (EMR) shielding in microwave range likewise for absorbing microwaves in radar operating range, as to act as radar absorbing materials (RAM). Consequently, the measurements of reflection and transmission of developed textile fabrics were performed in a frequency range of 2–18 GHz, which covers the defined frequencies relevant to the application. To improve the resistance performance of developed samples, a conventional textile surface modification technique, i.e., atmospheric plasma treatment, was used and the plasma effects were studied.

## 2. Materials and Methods

### 2.1. Materials and Their Treatment

Four groups of samples were manufactured for this research work where, for each group, different woven fabrics were used as the substrates, per Table 1. 

Plasma pre-treatment: To achieve a better resistance performance of coated fabrics, plasma treatment before coating was used. An industrial scale Atmospheric Plasma Treatment System from Sigma Technologies, model 2M (Sigma Technologies Int’l, LLC, Tucson AZ, 85737 USA), was used. Three types of treatment were used: oxygen (O_2_), nitrogen (N_2_), and Corona (air). For oxygen and nitrogen, 45% of these gasses was used, which correspond a flow rate of 450 sccm, and 100% of a carrier gas (argon), which correspond a flow rate of 15,000 sccm. The power of 15kW and speed of 3.3Hz (ca 10 m/min) was applied.

Before and after plasma treatment, the fabrics were evaluated regarding an aqueous liquid repellency test (according to ISO 23232 standard [36]), known as the drop test, and by measuring the contact angles for the water solvent, using the tensiometer equipment. The aqueous repellency grade/level is the highest numbered test liquid which is not absorbed by the substrate surface. The higher the aqueous solution repellency grade, the better the resistance to staining by aqueous materials, especially liquid aqueous substances. The aqueous solution repellency grade of a substrate is the numerical value of the highest-numbered test liquid which will not wet the substrate within a period of (10 ± 2) s. a grade of zero (0) is assigned when the substrate fails the 98% water solution test liquid.

According to the results obtained after preliminary experiment with different plasma treatments it was concluded that the plasma treatment with corona is the best, since there was a significant improvement in the hydrophilicity for those samples (S1 and S3), which initially were rather hydrophobic—their repellency level was 3, presenting a high hydrophobicity. For the other two fabrics (S2 and S4), the initial repellency level was 0 due to their high hydrophilicity. fabrics (substrates). Therefore, for selected fabrics—substrates (Table 1), before coating with conductive composition plasma treatment with corona (2 passages) was applied. After the optimization of plasma treatment, the fabrics were evaluated through the aqueous liquid repellence test. The results obtained to the untreated and plasma treated fabrics are presented in Table 2.

Coating: For imparting conductivity properties, samples of woven fabrics after plasma treatment as well as samples without plasma treatment were coated with the paste—Clevios SV3, produced by Heraeus (Hanau, Germany). Properties of conductive paste are presented in Table 3. The conductive composition used contains conjugated polymer system—poly(3,4-ethylenedioxythiophene)-polystyrene sulfonate (PEDOT:PSS), with concentration 1–3% *w*/*w*. The measured viscosity (20 °C) of Clevios SV3 is 20 Pa s and pH is 1.7. The weight ratio of PEDOT to PSS is about 1:2.5.

The tested substrates were coated in the laboratory, using a screen printing method. In order to bond and fix the conductive layer on the fabric, the samples were dried in laboratory oven and steamer TFOS IM 350 (Roaches International, Birstall, West Yorkshire, UK) at condensation temperature of 120 °C for 4 min.

### 2.2. Methods

#### 2.2.1. Structural Parameters 

Structural parameters of substrate fabrics were determined using standard test methods, respectively: mass per unit area (g/m^2^) according to EN 12127 [37]; determination of the thickness of textiles (mm) according EN ISO 5084 [38], number of threads per unit length according to EN 1049-2 [39].

The amount of coating deposit (g/m^2^) for coated samples was evaluated by measuring mass per unit area of manufactured coated samples and subtracting the value of mass per unit area of control sample.

#### 2.2.2. SEM Analysis

The surface morphology of the coated textile samples was examined applying scanning electron microscopy (SEM) and using a Quanta 200 FEG device (FEI, Eindhoven, Netherlands) at 20 keV (low vacuum). All microscopic images were made under the same technical and technological conditions: electron beam heating voltage—20.00 kV, beam spot—5.0, magnification—5000× and 10,000×, work distance—6.0 mm, low vacuum—80 Pa, and detector-LFD.

#### 2.2.3. Electrical Conductivity

In order to evaluate the electrical conductivity of samples the reverse value—surface resistance was measured according to EN 1149-1 [40] standard, with Terra-Ohm-Meter 6206 (produced by Eltex-Electrostatik-GmbH, Weil am Rhein, Germany) applying voltage of 10 V. Five test specimens were cut to a size between the overall dimensions of the electrodes and of the base plate from the material designed. Specimens were pressed with the load of about 10 N between an assembly of cylindrical and annular electrode arranged concentrically with each other and base plate, on which the specimen was placed. The diameter of electrode used—100 mm. The range of the ohmmeter used 10^3^ Ω to 10^14^ Ω. The conditioning for 24 h and tests were carried out in dry conditions—air temperature (23 ± 1) °C, relative humidity (25 ± 5) %, as indicated in the standard used. The measuring circuit of electrodes during measurements of resistance is presented in Figure 1.

The surface resistivity (*ρ*) was calculated according to equation:(1)ρ=k⋅R,
where *R* is the measured surface resistance, *k* is the geometrical factor of the electrode. The geometrical factor of the electrode is calculated according to equation:(2)$$k=\frac{2\cdot \pi }{\log_{e}\left(\frac{r_{2}}{r_{1}}\right)}=\frac{2\cdot 3.14}{\log_{e}\left(\frac{34.6}{25.2}\right)}=19.8,$$
where $r_{1}$ is the radius of the inner electrode and $r_{2}$ is the inner radius of the outer electrode.

#### 2.2.4. EMR Shielding, Reflection and Absorption 

Experimental investigations of reflection and transmission of electromagnetic waves normally incident on fabrics have been performed in the far-field area, using a semi anechoic chamber. The measurement setup is shown in Figure 2.

It is seen that the measured sample of the fabric is surrounded with absorber sheets preventing for the diffracted wave directly get to the receiving antenna. We used tuneable microwave generator as a microwave source. Continuous wave signal was passed to the transmitting antenna. Transmitted and reflected power was measured using average power sensors from Rhode & Schwarz (Fleet, Hampshire, UK) NRP-Z24. The maximum absolute uncertainty for power measurements for such a sensor is 0.222 dB, which is equal to ±5.24 %. The receiving antenna measuring reflection was positioned near the transmitting one. Direct coupling between them was less than −30 dB. The measured transmitted power *P_t_* with an object under the test was normalized by the power *P*_*t*0_ measured in the absence of it. Therefore, the transmittance can be expressed in the following way:(3)$$T=\frac{P_{t}}{P_{t0}}$$

Results of transmittance are presented in a form of EMR shielding effectiveness SE expressed in decibels:(4)$$SE=10\cdot \log_{10}\cdot T$$

In order to calculate the microwave reflectance from the fabric under the test, the reflected microwave power from the fabric *P_r_* is divided by the reflected power measured when the sample under the test is replaced with the same dimension metal plate *P_rm_*. Thus, the reflectance is calculated: (5)$$\Gamma =\frac{P_{r}}{P_{rm}}$$

Absorption in a tested sample has been determined as a difference between falling, transmitted and reflected waves:(6)A=1−T−Γ

Three sets of horn antennas for measurements in a frequency range 2.4–4.0 GHz (S-band WR284), 4–7.5 GHz (C band WR159) and 7.5–18.0 GHz (X-band WR90) were used in our experiments. Rohde & Schwarz average power sensor was connected to the receiving antennas using corresponding waveguide to coaxial line adaptor. Using our measurement setup EMR Shielding effectiveness of the order of 35 dB can be measured. Combined standard uncertainty for shielding effectiveness and reflection measurements is equal to ±7.4 % for a single data point at a fixed frequency.

#### 2.2.5. Calculation of the Surface Conductivity according to the Measurements of Reflectance and Transmittance

The fabrics covered with conductive paste can be considered as a thin layer with particular surface conductivity ***σ**_s_*** = 1/***R_s_***, were ***R_s_*** is surface resistance. Since the thickness of fabrics is much less than the wavelength of electromagnetic wave used in experiments, the dielectric properties of the fabric can be neglected. It also confirms our measurements of the substrate. We did not find any reflection or decrease of transmittance experimentally in the substrate fabric tests. 

We consider a plane electromagnetic wave incident normally on a surface of the conductive textile characterized by the only parameter—surface conductivity. A schematic view of the situation is shown in Figure 3.

On the left side we have incident and reflected waves whereas on the right side we have transmitted wave only. The problem can be solved analytically applying boundary conditions at the interface *z* = 0. It is clear that the tangential component of electric field should be continuous in the interface of two regions, whereas the magnetic field component should have a step due to surface conductivity [41].
(7)$$\left[n\times \left(H_{∥}\left(0_{+}\right)-H_{∥}\left(0_{-}\right)\right)\right]=J_{s}=\sigma_{s}\;E_{∥}\left(0\right)$$
where n is *z* axis directed unit vector perpendicular to the surface of the conductive layer, $H_{∥}$ (0_−_) and $H_{∥}$ (0_+_) are tangential components of the magnetic field at the left and right sides from the layer, J*_s_* is a surface current density, and $E_{∥}$ (0) is the tangential component of the electric field in the layer (bolded letters in (7) correspond to vectors).

Assuming that the amplitude of electric field of the incident wave is continuous, the amplitudes of electric and magnetic field in the first region can be written in the following way
(8)$$\vec{E}\left(z\right)={\vec{e}}_{x}\left[1e^{-ikz}+E^{\left(r\right)}e^{ikz}\right],$$
(9)$$\vec{H}\left(z\right)={\vec{e}}_{y}\left[\frac{1}{\eta }e^{-ikz}-\frac{E^{\left(r\right)}}{\eta }e^{ikz}\right],$$
where $k=\frac{2\pi }{\lambda }$ is a wavenumber and *η* is the impedance of free space. In the second region components of the electric and magnetic fields read: (10)$$\vec{E}\left(z\right)={\vec{e}}_{x}E^{\left(t\right)}e^{-ikz},$$
(11)$$\vec{H}\left(z\right)={\vec{e}}_{y}\frac{E^{\left(t\right)}}{\eta }e^{-ikz}.$$

The following boundary conditions should be satisfied at the interface
(12)$$E^{\left(I\right)}\left(0\right)=E^{\left(II\right)}\left(0\right),$$
(13)$$H^{I}\left(0\right)=H^{II}\left(0\right)+\sigma_{s}E^{II}\left(0\right).$$

Inserting there the expressions of amplitudes (8)–(11) one can get the system of two equations from which the reflection and transmission coefficients for amplitude can be obtained: (14)$$E^{\left(r\right)}=-\frac{\eta \sigma_{s}}{2+\eta \sigma_{s}},$$
(15)$$E^{\left(t\right)}=\frac{2}{2+\eta \sigma_{s}}.$$

As follows from the obtained expressions, reflection and transmission coefficients are real since the conductive sheet does not change a phase of reflected and transmitted waves. Reflectance and transmittance will be expressed as
(16)$$\Gamma ={\left[\frac{\eta \sigma_{s}}{2+\eta \sigma_{s}}\right]}^{2},$$
(17)$$T=\frac{4}{{\left(2+\eta \sigma_{s}\right)}^{2}}.$$

Absorption in the layer can be easily calculated multiplying the drop of the amplitude of magnetic field with the amplitude of electric field in the layer leading to
(18)$$A=\frac{4\eta \sigma_{s}}{{\left(2+\eta \sigma_{s}\right)}^{2}}.$$

It is easy to check that the sum of Γ, *T* and *A* equals to 1 that should follow from the energy conservation law. The obtained expressions (16), (17), and (18) can be used for the calculation of electromagnetic characteristics of the fabric samples covered with the conductive paste.

#### 2.2.6. Evaluation of Conductive Coating Durability on Washing 

Each fabric sample was subjecting to repeated 5 washing and drying cycles. The fabrics were washed in Scourotester Computex (Budapest, Hungary), using washing procedure described in EN ISO 105-C06 [42] standard, method A1S—washing temperature was 40 °C for 30 min with 150 mL of water containing 4 g/L of ECE Reference detergent with phosphates without optical brightener. Samples were then dried in ambient atmosphere.

#### 2.2.7. Evaluation of Conductive Coating Durability on Abrasion

Interfacial bonding strength was assessed using Martindale abrasion measurements. Abrasion tests were performed using a Nu-Martindale Abrasion and Pilling Tester (James Heal, Halifax, UK), according to EN ISO 12947-2 [43]. Samples were treated with standard wool abradant applying 9 kPa pressure and the change in appearance after 50, 100, 200, and 1000 cycles was evaluated visually.

## 3. Results and Discussion 

To develop the fabrics with EMR shielding and radar absorbing properties samples of woven fabrics were coated with the conductive paste—Clevios SV3 (Heraeus, Hanau, Germany), containing conjugated polymer system—poly(3,4-ethylenedioxythiophene)-polystyrene sulfonate (PEDOT:PSS). This paste was chosen not only for its merits mentioned in the introduction part, but also because of its suitability for screen printing or knife-over-roll coating technology, as our aim was to develop the fabric coated with conductive layer only on the back side, that it could be integrated in the military camouflage clothing system.

The substrate coated with conductive paste PEDOT:PSS and its yarn cross-section is illustrated in Figure 4. Analysis of SEM views of all coated substrates indicated that the thickness of the coating is not uniform and is approximately equal to 1 µm.

The most versatile form of PEDOT for processing is the synthesis of PEDOT as a polyelectrolyte complex [44]. The complex consists of polymeric cationic PEDOT and a polymeric counter anion [45]. The most effective counterion for PEDOT is polystyrenesulphonic acid (PSS) [8,19,45], which improves PEDOT solubility in water.

Researches assigned PEDOT:PSS to a “conducting acid dye” which can also tightly bind to protein fibres through electrostatic interaction of PSS chain negatively charged sulfonate (-SO3-) ions to protein fibre cationic sites [46]. The research of interaction between protein-based fibres and ICPs, PEDOT:PSS and poly(4-(2,3-dihydrothieno[3,4-b]-[1,4]dioxin-2-yl-methoxy)-1-butanesulfonic acid (PEDOT:S), at different pH was carried out in study [47]. The synthetic polyamide being the long chain polymers with recurring cationic amide (-CONH-) groups exhibited similar properties and can ionically bond as protein fibres [48]. Some trials to dye cotton yarns with PEDOT:PSS formulations were carried out by [49].

Therefore, for the investigated coated fabrics, the interaction between fibers and PEDOT:PSS happens likely due to electrostatic interaction among the water soluble conjugated polyelectrolyte PEDOT:PSS negatively charged sulfonate counterions with protonated amino groups in the wool and amide bonds in the polyamide as well as positively charged sites of chemically modified cotton.

However, the key obstacle for commercial application of conductive polymer coated fabrics is the poor adhesion of coating substance over the fiber surface [30,50]. To improve wear and washing resistance performance of fabrics coated with composition containing PEDOT:PSS, a conventional textile surface modification technique, atmospheric plasma treatment, was used.

As developed fabrics are intended for clothing protecting from EMR and against detection by battlefield radar, plasma treatment impact on their electrical and electromagnetic properties as well as on abrasion and washing resistance performance was studied.

### 3.1. Electrical Conductivity

The electrical conductivity of the material is one of the important factors influencing its EMR shielding. Usually, in order to evaluate the electrical conductivity of the textile material, inverse dimension—surface resistivity is measured. For determination of this parameter there was used standard test method EN 1149-1 [40], which is mostly applicable for textile materials intended for protective clothing. The surface resistivity (*ρ,* Ω) measurement results presented in Table 4 show that before washing all tested materials are possessed of near the same electrical conductivity, despite they are treated with plasma or not. After washing procedures, the surface resistivity of samples slightly increased, *ρ* increased by one row, which means that the conductivity rather decreased. In this case also there was no marked differences between plasma treated and untreated samples, maybe except for samples from PA/cotton fibers (S3C and S3PC), where a positive plasma treatment influence can be noticed. 

However, after analysis of the correlation between the surface resistivity of tested samples and EMR shielding effectiveness SE (dB) (Figure 5) was assessed, that knowledge of the surface resistivity, determined by the method used could not be used for the prediction of sample shielding ability. The coefficients of determination R^2^ = 0.2547 (Figure 5a) and R^2^ = 0.131 (Figure 5b) for the dependence of SE on the surface resistivity of samples before and after washing, respectively, shows that a correlation between these parameters is rather weak.

It stands to reason that results of *ρ* measurements, obtained for conductive paste coated samples with non-indiscrete surfaces (Figure 5), are not reliable and repeatable due to the method used, which does not exclude contact resistance in the system. The SEM images (Figure 6) illustrate that the surface of the coated side of fabric is not even and homogenous.

Therefore, obtained *ρ* values only show that applied conductive coating provides the fabrics with electrical properties, but does not allow objectively to distinguish the differences between samples and to predict their shielding ability.

### 3.2. Microwave Properties

The main purpose to coat fabrics with conductive paste was to provide them with the function to shield EMR in microwave range likewise to absorb microwaves in radar operating range.

A major threat to dismounted soldiers are battlefield radars commonly operating within X and Ku-bands at 8–12 GHz and 12–18 GHz, respectively [51]. Consequently, the investigation of reflection and transmission properties of developed textile fabrics was performed in a frequency range of 2–18 GHz, which cover the defined frequencies relevant to the application. The experimental data show the variation in EMR shielding effectiveness for four groups of coated fabrics over the tested range (Table 5 and Figure 7). For the uncovered fabrics (substrate samples), EMR shielding effectiveness SE (dB) was 0.

All samples were coated with conductive paste (Table 3) in the equal conditions: the same screen, two layers (with intermediate drying) and thermo-fixation at 120 °C for 4 min. To shape the first layer, 4 passages were used and for the second, 2. However, the coating deposits were obtained different for each group of samples due to their particular structure (Table 1) and surface properties (wettability, moisture absorption and transport). With reference to obtained results, it could be stated that depending on such fabric structural parameters as—density, mass per unit area (or just mass), type of weave, as well as the apertures (pores), layer of shield (or coating) just sticks on the fabric surface or penetrates into fabric changing the shield thickness herewith turning SE results.

Individually, for each type of fabric, the coating deposit and herewith the amount of conductive additive, PEDOT:PSS, have the most important impact on shielding properties. In our previous works [23], as well as during preliminary experiments for this study, SE determined as a function of coating deposit. For example, substrate S3 with increasing coating deposit—7 g/m^2^, 14 g/m^2^, 17 g/m^2^, 24 g/m^2^, demonstrated increased |SE|—10 dB, 15 dB, 20 dB, 25 dB, respectively. Similar correlation between coating deposit and shielding effectiveness was obtained for other fabrics under this study. For further investigations the fabrics with sufficiently high SE, but not exceeding 20 dB were selected, as it was found [5] that for radar absorbing only materials with total EMR SE below 20 dB were suitable.

The data presented in Table 5 reports the average (dominant) SE values of coated fabrics with and without plasma treatment. As seen from results (Table 5 and Figure 7) EMR shielding ability for plasma treated samples in each group remain almost the same as for samples without plasma treatment, maybe with exception of wool samples (samples S1C and S1PC). In the case of the last-mentioned samples, SE values for plasma treated sample (SIPC) is slightly higher in comparison with untreated sample (S1C) due to a bit increase in coating deposit. 

As can be seen from Figure 7, the shielding properties of coated samples are steady in all of the tested frequency range 2–18 GHz. Therefore, such a full side (in our case back side of the fabric) covered with conductive paste fabrics can be considered as a thin layer with particular surface conductivity.

The surface conductivity values calculated with reference to reflection and transmission measurements are presented in Table 5. It can be seen that calculated *σ* values strongly correlate with obtained SE values and allow to distinguish the samples with different conductivity, that is complicated using electrostatic parameter—surface resistivity, determined by the method conventional for textile testing.

To evaluate the ability of developed coated fabrics to absorb microwaves in radar operating frequency range, their reflection properties were measured (Figure 8) and absorption (A) coefficient calculated based on transmission and reflection measurements results (according to formulas (4) and (6)). The data presented in Table 6 show the contribution of reflection and absorption to the total EMR shielding effectiveness for each tested sample at 12 GHz and 18 GHz respectively.

As it is seen from Table 6, for tested samples the combined effect of reflection and absorption determines the shielding properties, but the role of these two parameters is different over the tested frequency range. As the reflection coefficient is more or less frequency depended for all tested samples (Figure 8), their reflectance properties are different over the 2–18 GHz range—reflectance decrease when the frequency increase. The lower reflection coefficients are obtained in the 12–18 GHz range, it means that in this range coated fabrics have the better absorption ability. Moreover, different types of fabric demonstrated different reflection and consequently absorption properties (Table 6). 

The best results were obtained for group II samples (Figure 9)—the contribution of absorption at 18 GHz sought about 52% and 74% respectively for sample without plasma treatment (S2C) and with plasma treatment (S2PC). For other tested fabrics, no significant difference in reflection and absorption between plasma treated and untreated sample was noticed.

### 3.3. Washing and Wearing Resistance Performance

To assess washing resistance of coated fabrics all tested samples after washing procedures were measured not only for surface resistivity (Table 4), but also for shielding effectiveness. SE measurement results before and after washing (Figure 10) showed that, for all 4 groups of tested samples, the resistance to washing increased for samples with plasma treatment in comparison with untreated samples. 

The samples after washing also were assessed visually. As an example, the photos of wool samples (I group) are presented in Figure 11.

Assessing the influence of washing on the durability of conductive coating, it could be stated that the plasma treatment slightly improved the durability of PEDOT:PSS coating.

The influence of plasma treatment on durability of coating was also observed after abrasion impact. As it can be seen from visual observations (Figure 12), the conductive coating survived after a longer period of abrasion cycles when samples were treated by plasma before coating procedure in comparison with untreated. These results clearly show that the plasma treatment was quite effective at improving the adhesion strength between textile substrate and the coating of a conductive composition containing PEDOT:PSS.

This change can be due to incorporation of additional functional groups formed by plasma reactions which changes the surface chemistry of textile substrate by decomposition of polymer chains and oxidation [50].

## 4. Conclusions

Different types of woven fabrics were coated using electrically conductive composition containing PEDOT:PSS. All samples developed for this study were characterized by surface resistivity and microwave properties within the 2–18 GHz frequency range. To improve the wear and washing resistance performance of the developed samples, conventional textile surface modification technique—atmospheric plasma treatment, was used.

Reflection and transmission measurements showed that for tested samples the combined effect of reflection and absorption determines the shielding properties, but the role of these two parameters is different over the tested frequency range. Shielding properties of investigated coated fabrics, with plasma treatment as well as without it, are steady in all tested frequency range and are sufficiently good for shielding/radar absorbing materials: SE for samples of groups I, II, and III is about 15–19 dB and for group IV 17–20 dB. There was no significant increase in SE due to plasma treatment, maybe except for samples of I group (wool fabric), where after the plasma treatment SE increased from 15 dB to 19 dB. This can be explained by the higher deposit of conductive paste on the fabric due to significant improvement in the hydrophilicity by plasma.

Meanwhile, reflection and consequently absorption are frequency-dependent. For all tested samples, the better absorption ability was obtained in the 12–18 GHz range. Moreover, it was noticed the positive effect of plasma treatment on absorption properties, particularly for samples of II (PA fabric) and III (PA/cotton) groups. The best results were obtained for II group’s samples (PA fabric): the contribution of absorption to shielding at 18 GHz was 52% for sample without plasma treatment and increased to 74% after plasma treatment.

It was found that EMR shielding effectiveness (SE) as well as absorption properties depend not only the amount of conductive paste topped on the fabric, but also resides in the construction parameters of fabrics and their finishing before coating. Depending on such fabric structural parameters as–density, mass per unit area, type of weave, layer of shield (or coating) just sticks on the fabric surface or penetrates into fabric changing the shield thickness herewith turning SE results. Meanwhile, the fiber composition of fabrics influences mostly bonding between fibers and polymer coating. The investigations have shown that applied plasma treatment visibly improved abrasion resistance as a result of better adhesion between textile substrate and coating. The most effectively plasma influenced abrasion resistance of wool (I group) and PA/cotton (III group) fabrics. SE measurement results before and after washing procedures showed that plasma treatment somewhat improved washing resistance of conductive coat. Comparing the shielding effectiveness of washed coated fabrics without and with plasma treatment, it was determined that for all tested fabrics with plasma treatment SE decreases less liken to untreated fabrics: for group I (wool fabrics) SE decreases 3.8 times for the plasma treated sample and 7.5 times for untreated and for other groups respectively, for group II (PA fabrics) 2.8 and 7.2 times; for group III (PA/cotton fabrics) 2.5 and four times; for group IV (aramid/viscose fabrics) 5.6 and eight times. However, washing resistance increased not so considerably as the binding strength of the coating was insufficient for such impact. A more significant improvement of resistance to washing of such coated fabrics will be the subject of our subsequent study.

## Figures and Tables

**Figure 1 polymers-13-01191-f001:**
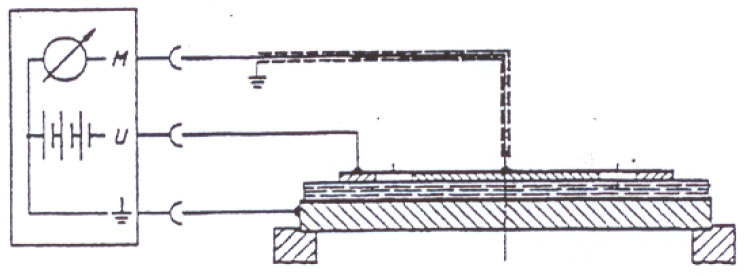
Measuring circuit of surface resistance measurement.

**Figure 2 polymers-13-01191-f002:**
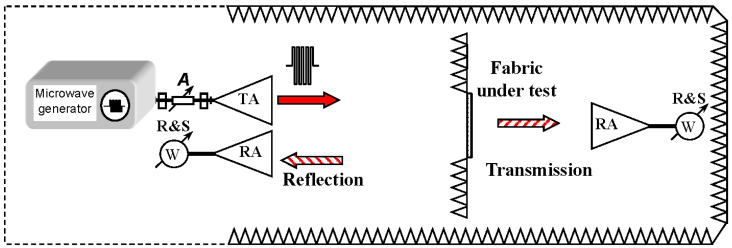
Measurement setup of microwave transmission and reflection from the coated fabric. TA denotes transmitting and RA—receiving antennas.

**Figure 3 polymers-13-01191-f003:**
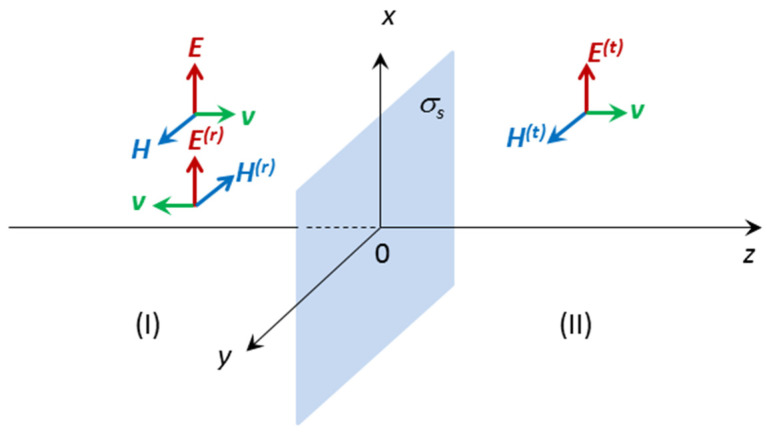
The model of fabric covered with conductive paste. On the left side of the sample there are incident and reflected waves. The later is denoted by superscript (*r*). On the right side the only transmitted wave appears. Its amplitudes are denoted by superscript (*t*).

**Figure 4 polymers-13-01191-f004:**
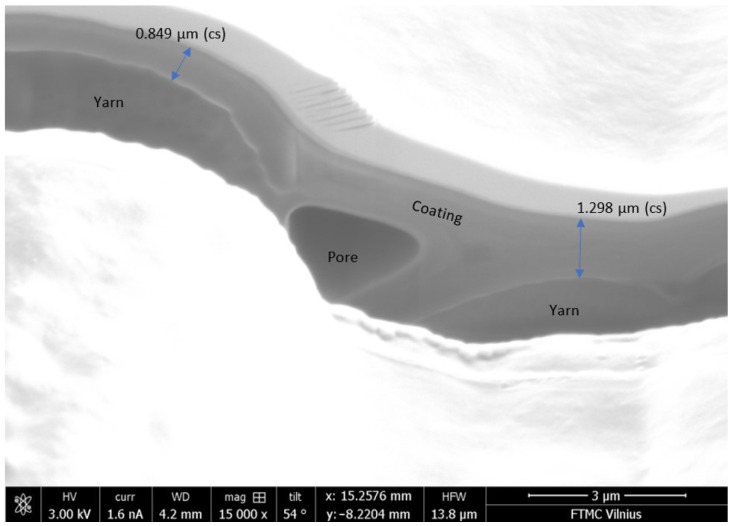
SEM image of S3C sample coated with PEDOT:PSS paste.

**Figure 5 polymers-13-01191-f005:**
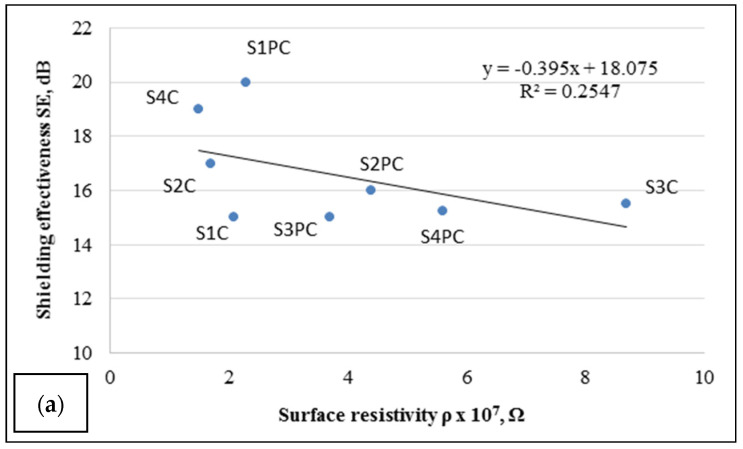
Dependence of surface resistivity on shielding effectiveness: (**a**) unwashed samples; (**b**) samples after washing.

**Figure 6 polymers-13-01191-f006:**
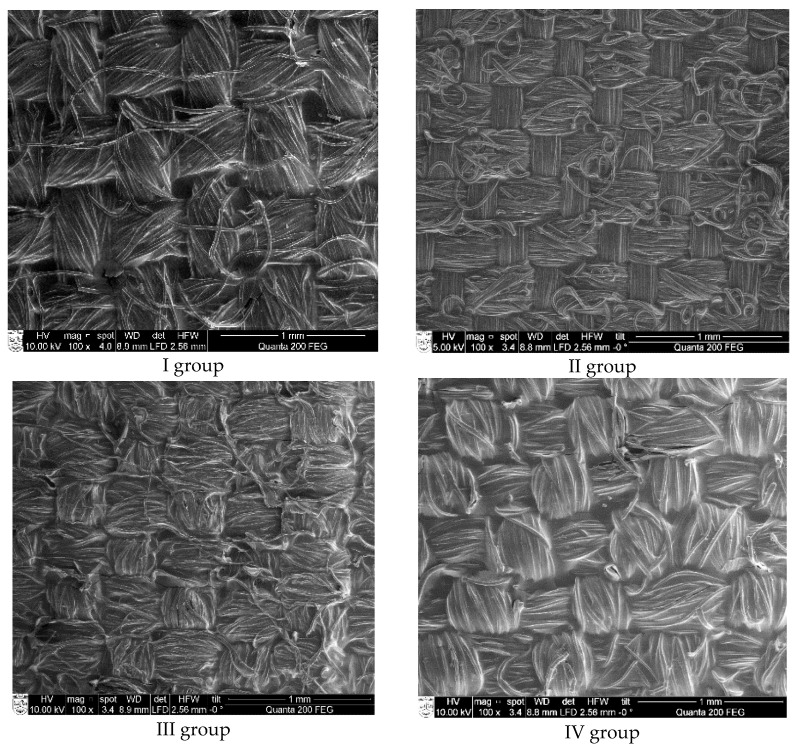
SEM images of samples coated with conductive paste.

**Figure 7 polymers-13-01191-f007:**
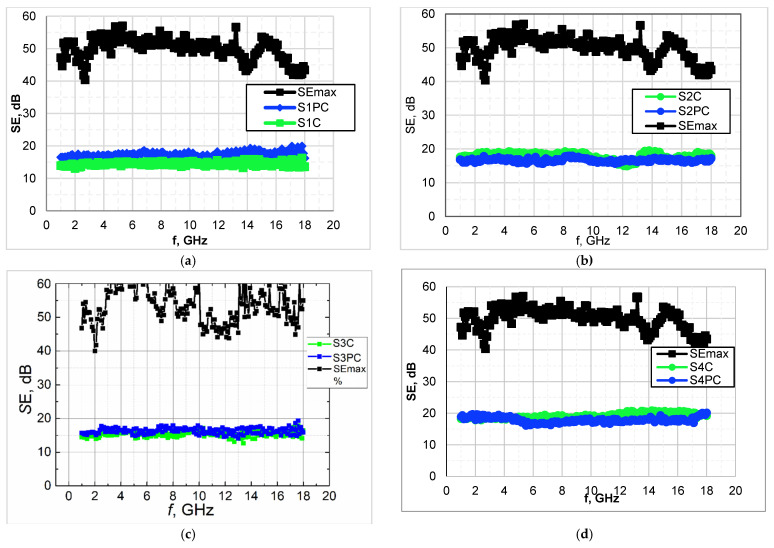
Shielding effectiveness SE, dB of tested fabrics coated with PEDOT:PSS formulation: (**a**)—I group of samples; (**b**)—II group of samples; (**c**)—III group of samples; (**d**)—IV group of samples.

**Figure 8 polymers-13-01191-f008:**
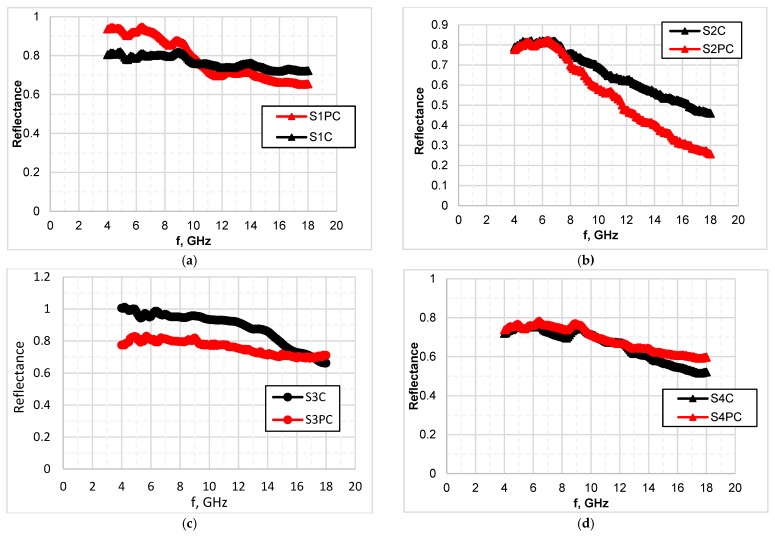
Reflectance of tested fabrics coated with PEDOT:PSS formulation: (**a**)—I group of samples; (**b**)—II group of samples; (**c**)—III group of samples; (**d**)—IV group of samples.

**Figure 9 polymers-13-01191-f009:**
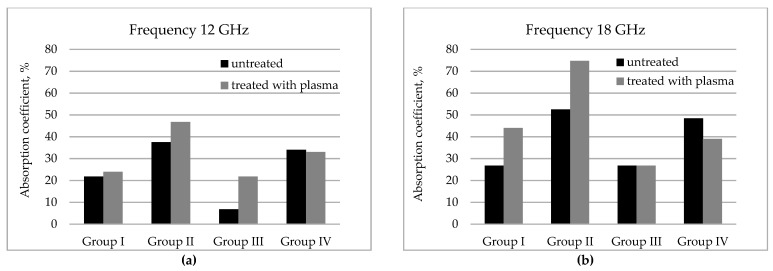
Absorption ability of tested samples: (**a**)—at 12 GHz frequency and (**b**)—at 18 GHz frequency.

**Figure 10 polymers-13-01191-f010:**
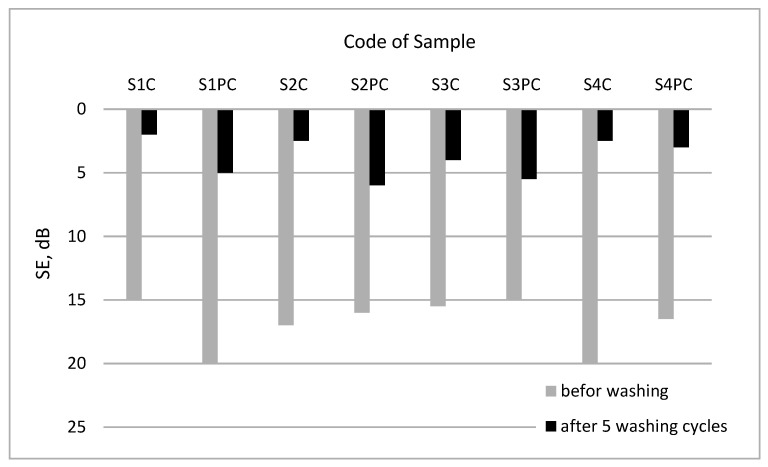
Influence of plasma treatment of shielding effectiveness before and after washing procedure.

**Figure 11 polymers-13-01191-f011:**
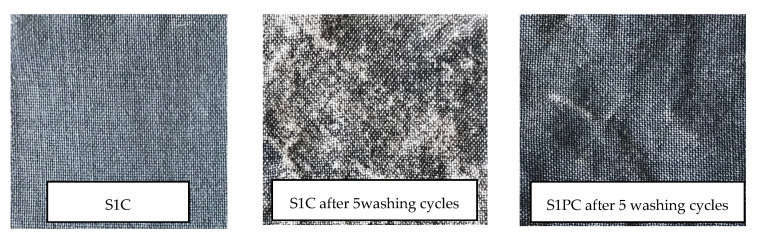
An appearance of coated side of fabric after washing procedures.

**Figure 12 polymers-13-01191-f012:**
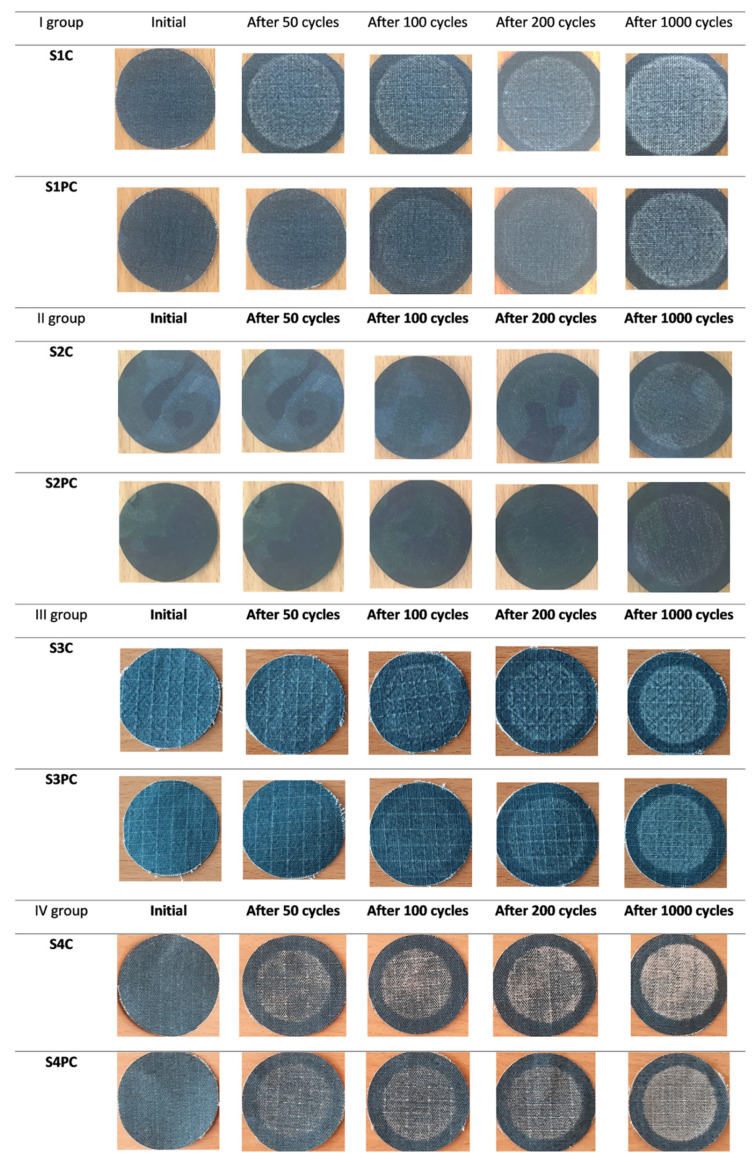
Coated fabric samples after being subjected to abrasion in the Martindale abrasion tester.

**Table 1 polymers-13-01191-t001:** The description of fabrics developed for the investigation.

Group of Samples (and Substrate View, Magnified ×60)		Code of Fabric	Description	Mean Mass Per Unit Area *, g/m^2^	Thickness, mm	Number of Threads Per Unit Length	
						No. of Warp Threads Per 1 cm	No. of Weft Threads Per 1 cm
**I**	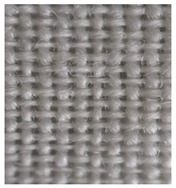	S1	Substrate (1)—100 % wool woven fabric (without plasma treatment and without conductive coating)	127	0.49	22	18
		S1C	Substrate (1) coated with conductive paste				
		S1PC	Substrate (1) treated with plasma and coated with conductive paste				
**II**	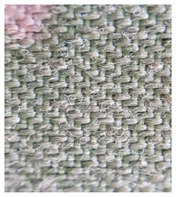	S2	Substrate (2)—100 % polyamide woven fabric, printed with camouflage pattern (without plasma treatment and without conductive coating)	130	0.45	47	32
		S2C	Substrate (2) coated with conductive paste				
		S2PC	Substrate (2) treated with plasma and coated with conductive paste				
**III**	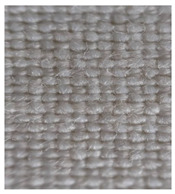	S3	Substrate (3)—58 % cotton/42% polyamide woven fabric (without plasma treatment and without conductive coating)	227	0.67	28	25
		S3C	Substrate (3) coated with conductive paste				
		S3PC	Substrate (3) treated with plasma and coated with conductive paste				
**IV**	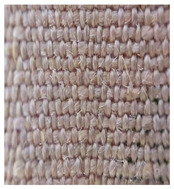	S4	Substrate (4)—Aramid/viscose 55%/45% woven fabric, printed with camouflage pattern (without plasma treatment and without conductive coating)	236	0.67	38	24
		S4C	Substrate (4) coated with conductive paste				
		S4PC	Substrate (4) treated with plasma and coated with conductive paste				

*—The evaluated expanded uncertainties (U) for measurement of mean mass per unit area is ±5 g/m^2^. The reported expanded uncertainties are based on a standard uncertainty multiplied by a coverage factor k = 2, which for a normal distribution provides a level of confidence of approximately 95%.

**Table 2 polymers-13-01191-t002:** Level of aqueous liquid repellence test for the untreated and treated fabrics with corona, after one and two passages.

Code of Fabric	No Treatment	Treatment (1 Passage)	Treatment (2 Passages)
S1	Level 3	Level 1	Level 1
S2	Level 0	Level 0	Level 0
S3	Level 3	Level 1	Level 0
S4	Level 0	Level 0	Level 0

**Table 3 polymers-13-01191-t003:** Properties of conductive paste applied (product supplier information)

Properties	
Composition/information on ingredients (component name and % by weight)	Propylene glycol ≤ 66.5 2,2′-oxydiethanol ≤ 15 Benzensulphonic Acid, Ethenyl-, Homopolymer, Compd.with 2,3-dihydrothienol[3,4-b]-1,4-dioxin Homopolymer ≤ 15
Surface resistivity (test print)	700 Ω/sq
Product description (supplied form)	Aqueous dispersion

**Table 4 polymers-13-01191-t004:** The electrostatic properties of investigated fabrics.

Group of Samples	Code of Sample	Surface Resistivity * *ρ*, Ω	Surface Resistivity *After Washing *ρ*, Ω
**I**	S1C	2.1 × 10^7^	1.7 × 10^8^
	S1PC	2.3 × 10^7^	2.6 × 10^8^
**II**	S2C	1.7 × 10^7^	4.3 × 10^8^
	S2PC	4.4 × 10^7^	3.3 × 10^8^
**III**	S3C	8.7 × 10^7^	2.1 × 10^8^
	S3PC	3.7 × 10^7^	4.5 × 10^8^
**IV**	S4C	1.5 × 10^7^	1.0 × 10^8^
	S4PC	5.6 × 10^7^	3.7 × 10^8^

***** The evaluated expanded uncertainty (U) for surface resistivity (*ρ*) measurement U = *ρ*·0.12 [Ω]; The reported expanded uncertainties are based on a standard uncertainty multiplied by a coverage factor k = 2, which for a normal distribution provides a level of confidence of approximately 95%.

**Table 5 polymers-13-01191-t005:** EMR shielding effectiveness of investigated fabrics.

Group of Samples	Code of Sample	Dominant Shielding Effectiveness SE, dB (2–18 GHz)	Deposit of Conductive Paste, g/m^2^	Surface Conductivity, S
**I**	S1C	15	10	2.5 × 10^−2^
	S1PC	18–19	12	4.0 × 10^−2^
**II**	S2C	16–18	5	3.5 × 10^−2^
	S2PC	15–17	5	3.2 × 10^−2^
**III**	S3C	15–16	14	2.5 × 10^−2^
	S3PC	15	15	2.5 × 10^−2^
**IV**	S4C	18–20	6	4.7 × 10^−2^
	S4PC	17–20	7	4.7 × 10^−2^

**Table 6 polymers-13-01191-t006:** Microwave properties of tested coated fabrics.

Group of Samples	Code of Sample	Frequency					
		12 GHz			18 GHz		
		Components of Shielding Effectiveness %					
		Γ	A	T	Γ	A	T
**I**	S1C	75.0	21.8	3.2	70.0	26.8	3.2
	S1PC	75.0	24.0	1.0	55.0	44.0	1.0
**II**	S2C	60.0	37.5	2.5	45.0	52.5	2.5
	S2PC	50.0	46.8	3.2	25.0	74.8	2.0
**III**	S3C	90.0	6.8	3.2	70.0	26.8	3.2
	S3PC	75.0	21.8	3.2	70.0	26.8	3.2
**IV**	S4C	65.0	34.0	1.0	50.0	48.4	1.6
	S4PC	65.0	33.0	2.0	60.0	39.0	0.1

## Data Availability

The data presented in this study are available on request from the corresponding author.

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
