# Peer review of "Development and Investigation of PEDOT:PSS Composition Coated Fabrics Intended for Microwave Shielding and Absorption"

_polymers, 2021, doi:10.3390/polym13081191_

Round 1

Reviewer 1 Report

The manuscript by Rubeziene et al. presents an interesting study on the preparation of PEDOT:PSS coated fabrics for microwave shielding and absorption. The authors used various characterization techniques to examine the different fiber compositions and their influence on the EMR shielding effectiveness. The investigations also revealed that the plasma treatment improved abrasion resistance and longevity of device. The results shed light on potential applications as EMR shielding materials. The experimental results and interpretations are sound. And thus I recommend publishing this manuscript. The specific comments are as follows:

1) What is the viscosity and PH value of the conductive paste - Clevios SV3? Is the paste acidic or neutral? 

2) One of the merits of a conductive polymer is to obtain the flexibility of the device (ie. flexible electronics). It should be good idea if the merits are addressed in the introduction section. 

3) Could author explain the effect of PEDOT:PSS loading(wt% in textile) on the EMR shielding performance?

4) I suggest authors to include SEM image of fiber cross-section and indicate the thickness of PEDOT:PSS on fiber surface.

5) Page 3, Line 108. "The most common techniques for applying ICPs on textiles are solution coating, in situ polymerization, polymerization in supercritical fluid, electrochemical polymerization, electrostatic spinning, coating by screen printing, inkjet printing or knife-over-roll technology." I recommend authors to include a reference for each preparation techniques. I listed a few paper below you can use.

    a) in situ polymerization

    Liu, et al. "The effect of in-situ polymerization on PEDOT-PSS/PAN composite conductive fiber." IOP Conference Series: Earth and Environmental Science. Vol. 218. No. 1. IOP Publishing, 2019.

    b) electro-spinning

    Ding, et al. "Scalable and facile preparation of highly stretchable electrospun PEDOT: PSS@ PU fibrous nonwovens toward wearable conductive textile applications." ACS applied materials & interfaces 9.35 (2017): 30014-30023.

    c) screen printing/inkjet printing

    Guo, et al. "PEDOT: PSS “wires” printed on textile for wearable electronics." ACS applied materials & interfaces 8.40 (2016): 26998-27005.

    d) solution coating

        Otley, et al. "Phase segregation of PEDOT: PSS on textile to produce materials of> 10 A mm− 2 current carrying capacity." Macromolecular Materials and Engineering 302.3 (2017): 1600348.

6)  I suggest authors to report the PEDOT to PSS ratio used in the experiment.

7)  For the PEDOT:PSS impregnated textile fibers, there could be possible surface interaction happening between PSS and fiber surface, which contributed to the improved EMR shielding performance. I suggest authors to include a brief discussion on this part. I have included a paper below you can check on (in terms of PEDOT:PSS phase segragation on fiber surface).

Ref: Otley, et al. "Phase segregation of PEDOT: PSS on textile to produce materials of> 10 A mm− 2 current carrying capacity." Macromolecular Materials and Engineering 302.3 (2017): 1600348.

Author Response

Response to Reviewer 1 Comments

Point 1: What is the viscosity and PH value of the conductive paste - Clevios SV3? Is the paste acidic or neutral? 

 Response 1: According to suppliers’ information the viscosity of the paste used is between 1.5 to 20 Pa s. We experimentally found it equal to 20 Pa s (at 20 C). Also, this paste is acidic (measured pH is 1.7). The text is supplemented by this information.

Point 2: One of the merits of a conductive polymer is to obtain the flexibility of the device (ie. flexible electronics). It should be good idea if the merits are addressed in the introduction section.

Response 2: More merits of PEDOT:PSS are supplemented in Introduction part, besides the fact that the coating does not affect the mechanical properties of the substrate allows for them to be used as flexible and deformable substrates.

Point 3: Could author explain the effect of PEDOT:PSS loading(wt% in textile) on the EMR shielding performance?

Response 3: The amount of conductive additives, herewith PEDOT:PSS loading (wt% in textile), significantly influences shielding effectiveness – SE. Increasing an amount of conductive additive surface conductivity is on the rise that lead to an enhancement of SE (please see formula 4 in revised manuscript).

Point 4: I suggest authors to include SEM image of fiber cross-section and indicate the thickness of PEDOT:PSS on fiber surface.

Response 4: The suggestion is accepted: SEM images with fiber cross-section as also the thickness of paste on fiber surface now are presented in the text (Results and Discussion part).

Point 5: Page 3, Line 108. "The most common techniques for applying ICPs on textiles are solution coating, in situ polymerization, polymerization in supercritical fluid, electrochemical polymerization, electrostatic spinning, coating by screen printing, inkjet or knife-over-roll technology." I recommend authors to include a reference for each preparation techniques. I listed a few paper below you can use.

  1.   a) in situ polymerization

    Liu, et al. "The effect of in-situ polymerization on PEDOT-PSS/PAN composite conductive fiber." IOP Conference Series: Earth and Environmental Science. Vol. 218. No. 1. IOP Publishing, 2019.

  1.   b) electro-spinning

    Ding, et al. "Scalable and facile preparation of highly stretchable electrospun PEDOT: PSS@ PU fibrous nonwovens toward wearable conductive textile applications." ACS applied materials & interfaces 9.35 (2017): 30014-30023.

  1.   c) screen printing/inkjet printing

    Guo, et al. "PEDOT: PSS “wires” printed on textile for wearable electronics." ACS applied materials & interfaces 8.40 (2016): 26998-27005.

  1.   d) solution coating

        Otley, et al. "Phase segregation of PEDOT: PSS on textile to produce materials of> 10 A mm− 2 current carrying capacity." Macromolecular Materials and Engineering 302.3 (2017): 1600348.

Response 5: The recommendation is fulfilled – references for each preparation techniques are indicated in Introduction part. Citations are added to mentioned paragraph.

Point 6:  I suggest authors to report the PEDOT to PSS ratio used in the experiment.

Response 6: The suggestion is accepted: The weight ratio of PEDOT to PSS is about 1:2.5 and now are presented in the text (Experimental part).

Point 7:  For the PEDOT:PSS impregnated textile fibers, there could be possible surface interaction happening between PSS and fiber surface, which contributed to the improved EMR shielding performance. I suggest authors to include a brief discussion on this part. I have included a paper below you can check on (in terms of PEDOT:PSS phase segragation on fiber surface).

Ref: Otley, et al. "Phase segregation of PEDOT: PSS on textile to produce materials of> 10 A mm− 2 current carrying capacity." Macromolecular Materials and Engineering 302.3 (2017): 1600348.

Response 7: Results and Discussion part is supplemented with explanation: To develop the fabrics with EMR shielding and radar absorbing properties samples of woven fabrics were coated with the conductive paste – Clevios SV3, containing conjugated polymer system – poly(3,4-ethylenedioxythiophene)-polystyrene sulfonate (PEDOT-PSS). This paste was chosen not only for its merits mentioned in the introduction part, but also because of its suitability for screen printing or knife-over-roll coating technology, as our aim was to develop the fabric coated with conductive layer only on the back side, that it could be integrated in the military camouflage clothing system.

The most versatile form of PEDOT for processing is the synthesis of PEDOT as a polyelectrolyte complex [40]. The complex consists of polymeric cationic PEDOT and a polymeric counter anion [41]. The most effective counterion for PEDOT is polystyrenesulphonic acid (PSS) [8, 41, 42], which improves PEDOT solubility in water.

Researches assigned PEDOT:PSS to a “conducting acid dye” which can also tightly bind to protein fibres through electrostatic interaction of PSS chain negatively charged sulfonate (-SO3-) ions to protein fibre cationic sites [43]. The research of interaction between protein-based fibres and ICPs, PEDOT:PSS and poly(4-(2,3-dihydrothieno[3,4-b]-[1,4]dioxin-2-yl-methoxy)-1-butanesulfonic acid (PEDOT:S), at different pH was carried out in study [44]. The synthetic polyamide being the long chain polymers with recurring cationic amide (-CONH-) groups exhibited similar properties and can ionically bond as protein fibres [45]. Some trials to dye cotton yarns with PEDOT:PSS formulations were carried out by [46].

Therefore for investigated coated fabrics the interaction between fibers and PEDOT:PSS happens likely due to electrostatic interaction among the water soluble conjugated polyelectrolyte PEDOT:PSS negatively charged sulfonate counterions with protonated amino groups in the wool and amide bonds in the polyamide as well as positively charged sites of chemically modified cotton.

Reviewer 2 Report

Below are my in-depth comments to the article:

The Intruduction section is relatively extensive, but does not contain references to the literature in essential elements. In particular, the following lines require citation from specialized literature: from 48 to 59, from 80 to 84 and from 108 to 118.

The composition of the article is also relatively strange and should be improved. Firstly, the appearance of the samples after washing (Fig. 9) and the abrasive treatment (Table 7)  should appear earlier - probably in the materials and methods section. Similarly, it is surprising that the purely theoretical considerations related to Figure 5, its description and equations (7) to (18) appear in the results and discussion section. These are theoretical issues related to the measurement and should appear in the method section when discussing the measurement method of transmission, reflection and absorption.

Another controversial issue is the assumed SE level from which it makes sense to analyze the reflection and absorption components. One can imagine a situation where the SE value will be relatively low (high transmission), but the absorption share will be relatively large as there will be a small reflection. Then, although the shield has a high absorption component, it will actually be a weak shield. Therefore, it seems to me that transmission (or SE), reflection and absorption should be shown in the graphs (Fig. 6 and Fig. 7) using a measure that allows their comparison: e.g. in dB or as a percentage value (especially that the authors in line 421 refer to percentages that the reader cannot find in the charts).

The conclusions are too general and somehow detached from the article. The article contains interesting research material and relatively well-conducted measurements, while the conclusions section does not reflect this. Please work on this section to highlight interesting and original elements.

I am also interested in how the accuracy of the applied measurement method of transmission and reflection can be determined? With what error are the measurements made?

I also miss a comparison of the screening effectiveness of similar solutions with the use of other conductive polymers, such as PANI or PPy? In my opinion, the PEDOT-PSS shows the worst conductivity and thus the worst shielding efficiency, which is not confirmed in Table 8, where the SE is at a decent level of about 16dB. Of course, it all depends on the number of layers and thickness of coatings of this polymer.

It is also surprising in Table 8 that the use of plasma treatment does so little to improve SE!

Author Response

Response to Reviewer 2 Comments

Point 1: The Intruduction section is relatively extensive, but does not contain references to the literature in essential elements. In particular, the following lines require citation from specialized literature: from 48 to 59, from 80 to 84 and from 108 to 118.

 Response 1: The recommendation is fulfilled – citations are added to mentioned paragraphs (due to corrections line numbers has changed).

Point 2: The composition of the article is also relatively strange and should be improved. Firstly, the appearance of the samples after washing (Fig. 9) and the abrasive treatment (Table 7)  should appear earlier - probably in the materials and methods section. Similarly, it is surprising that the purely theoretical considerations related to Figure 5, its description and equations (7) to (18) appear in the results and discussion section. These are theoretical issues related to the measurement and should appear in the method section when discussing the measurement method of transmission, reflection and absorption.

Response 2: The theoretical consideration related to Figure 5, its description and equations (7) to (18) are moved to Experimental part. The washing and abrasion treatment methodologies are given in 2.2 part, however the Figure 9 and Table 7 presents our obtained results related to plasma treatment effect on washing and wear resistance performance, and their sutability for the result part is substantiated.

Point 3: Another controversial issue is the assumed SE level from which it makes sense to analyze the reflection and absorption components. One can imagine a situation where the SE value will be relatively low (high transmission), but the absorption share will be relatively large as there will be a small reflection. Then, although the shield has a high absorption component, it will actually be a weak shield. Therefore, it seems to me that transmission (or SE), reflection and absorption should be shown in the graphs (Fig. 6 and Fig. 7) using a measure that allows their comparison: e.g. in dB or as a percentage value (especially that the authors in lin 421 refer to percentages that the reader cannot find in the charts).

Response 3: The three percentage components of SE are presented as a result of measurements (reflection and transmition) and calculations (absorption) (revised version of manuscript Table 6).

Point 4: The conclusions are too general and somehow detached from the article. The article contains interesting research material and relatively well-conducted measurements, while the conclusions section does not reflect this. Please work on this section to highlight interesting and original elements.

Response 4: The conclusions are improved. The corrected version is presented in revised manuscript.

Point 5: I am also interested in how the accuracy of the applied measurement method of transmission and reflection can be determined? With what error are the measurements made?

Response 5: In our measurement, we used the R&S NRP-Z24 power sensor which has maximum absolute uncertainty of 0.222 dB (±5.24 %) and typical noise of 13 nW. In the case of high shielding effectiveness (SE) which is close to the maximum SE (40-50 dB), that can be measured, this noise can have a large influence on measurement accuracy. In our case, the SE of tested fabrics was around 20 dB and the noise influence can be neglected. In order to measure the transmittance or reflectance of the sample the power should be measured twice and because of that the combined standard uncertainty for these measurements is ±7.4 % for a single data point at a fixed frequency. Because of that, some fluctuations can be observed in SE and reflectance measurement results. If we assume that the SE of the tested fabric doesn’t depend on frequency (as in the case of S4C fabric), then we can calculate the mean square error which in the case of S4C fabric is equal to ±2.8%.

Point 6: I also miss a comparison of the screening effectiveness of similar solutions with the use of other conductive polymers, such as PANI or PPy? In my opinion, the PEDOT-PSS shows the worst conductivity and thus the worst shielding efficiency, which is not confirmed in Table 8, where the SE is at a decent level of about 16dB. Of course, it all depends on the number of layers and thickness of coatings of this polymer.

Response 6: As it was indicated in the Introduction part (from 143 to 147 - original version) the aim of this study was to develop EMR shielding/radar absorbing textile materials. The desired property of EMR shielding materials is low transmission or high shielding effectiveness (SE). This however, does not imply good absorption properties, leading to a reduction of the reflection which is important for radar protection. A structure with low transmission can still be highly reflective. Consequently, in order to be effective for a radar signature reduction application, the conductivity must not be too high since such a textile material would be too reflective, resulting in poor radar protection properties (also see introduction part lines 73-79 ). PEDOT-PSS was chosen for this study because of its processability, stable electrical conductivity and low price in comparison with other ICPs (this is indicated in the introduction part lines 114-116). The other merits of PEDOT:PSS are supplemented in the introduction part.

Point 7: It is also surprising in Table 8 that the use of plasma treatment does so little to improve SE!

Response 7: Plasma treatment was used to ensure better wear and wash resistance performance of coated materials. In this case SE improvement was not aimed at because technological parameters allow the regulation and acquisition of the SE to be achieved. Also please see Response 6.

Reviewer 3 Report

What does the hydrophobicity level of 0-4 mean? It should be clarified in the text.

What do the authors mean by “resistance performance of fabrics” in line 276? It is not related to the previous sentence. Similarly, in line 426, the term “washing resistance” should be replaced with “washing fastness”.

The results show that the plasma treatment did not have any effect in increasing the shielding effectiveness. So, it should be explained clearly in the text, what was the reason of selecting this approach?

What were the thicknesses of the applied coatings?

The details of coating formulations and the content of each ingredient should be added.

The novelty of the work and the differences with similar research should be mentioned in the introduction part. Relevant literature in this area should be cited. For instance:

  • Advances in Colloid and Interface Science (2020), 277, 102116.
  • Composites Science and Technology (2020) 195, 108186.
  • Fibers and Polymers (2015) 16(3), 585-592.
  • Chemical Engineering Journal, (2016), 297, 170-179.
  • Materials Research Bulletin, (2016) 83, 96-107.

Author Response

Response to Reviewer 3 Comments

Point 1: What does the hydrophobicity level of 0-4 mean? It should be clarified in the text.

Response 1: The text is supplemented by this information: The aqueous repellency grade/level is the highest numbered test liquid which is not absorbed by the substrate surface. The higher the aqueous solution repellency grade, the better the resistance to staining by aqueous materials, especially liquid aqueous substances. The aqueous solution repellency grade of a substrate is the numerical value of the highest-numbered test liquid which will not wet the substrate within a period of (10±2) s. a grade of zero (0) is assigned when the substrate fails the 98 % water solution test liquid.

Point 2: What do the authors mean by “resistance performance of fabrics” in line 276? It is not related to the previous sentence. Similarly, in line 426, the term “washing resistance” should be replaced with “washing fastness”.

Response 2: In Introduction part there is given explanation that resistance performance of fabrics mean that fabrics are resistance eg., to washing, withstanding other impacts, which appear during the wear. It is not related to electrical resistance. We tried to add wash and wear terms to the text to avoid misunderstanding.

Point 3: The results show that the plasma treatment did not have any effect in increasing the shielding effectiveness. So, it should be explained clearly in the text, what was the reason of selecting this approach?

Response 3: Plasma treatment was used to ensure better wear and wash resistance performance of coated materials. In this case SE improvement was not aimed at because technological parameters allow the regulation and acquisition of the SE to be achieved.

Point 4: What were the thicknesses of the applied coatings?

Response 4: Replaying to your comment we added SEM image with fiber cross-section in which the thickness of applied coating is shown.

Point 5: The details of coating formulations and the content of each ingredient should be added.

Response 5: Information on ingredients (component name and % by weight) are added to experimental part (text and table 3).

Point 6: The novelty of the work and the differences with similar research should be mentioned in the introduction part. Relevant literature in this area should be cited. For instance:

  • Advances in Colloid and Interface Science (2020), 277, 102116.
  • Composites Science and Technology (2020) 195, 108186.
  • Fibers and Polymers (2015) 16(3), 585-592.
  • Chemical Engineering Journal, (2016), 297, 170-179.
  • Materials Research Bulletin, (2016) 83, 96-107.

Response 6: The novelty of the research was highlighted. Some points added to Introduction part.